# Prognostic Value of Tpeak–Tend Interval in Early Diagnosis of Duchenne Muscular Dystrophy Cardiomyopathy

**DOI:** 10.3390/diagnostics13142381

**Published:** 2023-07-15

**Authors:** Serra Baskan, Pelin Karaca Ozer, Huseyin Orta, Doruk Ozbingol, Mustafa Lutfi Yavuz, Elif Ayduk Govdeli, Kemal Nisli, Kazim Oztarhan

**Affiliations:** 1Department of Pediatric Cardiology, Istanbul Faculty of Medicine, Istanbul University, Istanbul 34134, Turkey; skaraca92@gmail.com (S.B.); dorukozbingol@gmail.com (D.O.); kemal.nisli@istanbul.edu.tr (K.N.); kazimoztarhan@yahoo.com (K.O.); 2Department of Cardiology, Istanbul Faculty of Medicine, Istanbul University, Istanbul 34134, Turkey; huseyinorta15@gmail.com (H.O.); mustafayavuz135@gmail.com (M.L.Y.); elifayduk@gmail.com (E.A.G.)

**Keywords:** Duchenne muscular dystrophy, premature ventricular contraction, Tpeak–Tend interval

## Abstract

The most common cause of death in patients with Duchenne muscular dystrophy (DMD) is cardiomyopathy. Our aim was to investigate the relationship between the Tpeak–Tend (Tp-e) interval and the premature ventricular contraction (PVC) burden and therefore early arrhythmic risk and cardiac involvement in DMD patients. Twenty-five patients with DMD followed by pediatric cardiology were included in the study. Those with a frequency of <1% PVC in the 24 h Holter were assigned to Group 1 (*n* = 15), and those with >1% were assigned to Group 2 (*n* = 10). Comparisons were made with healthy controls (*n* = 27). Left ventricular ejection fraction (LVEF) was lowest in Group 2 and highest in the control group (*p* < 0.001). LV end-diastolic diameter was greater in Group 2 than in Group 1 and the control group (*p* = 0.005). Pro-BNP and troponin levels were higher in Group 1 and Group 2 than in the control group (*p* = 0.001 and *p* < 0.001, respectively). Tp-e interval was longer in Group 2 compared to Group 1 and the control group (*p* < 0.001). The LVEF (OR 0.879, 95% CI 0.812–0.953; *p* = 0.002) and Tp-e interval (OR 1.181, 95% CI 1.047–1.332; *p* = 0.007) were independent predictors of PVC/24 h frequency of >1%. A Tp-e interval > 71.65 ms predicts PVC > 1%, with a sensitivity of 80% and a specificity of 90% (AUC = 0.842, 95% CI (0.663–1.000), *p* = 0.001). Determination of Tp-e prolongation from ECG data may help in the determination of cardiac involvement and early diagnosis of arrhythmic risk in DMD.

## 1. Introduction

Duchenne muscular dystrophy (DMD) is a severe, progressive, X-linked genetic disease that affects roughly one in every 5000 live male births. This progressive disease causes severe clinical symptoms; progressive muscle weakness typically begins to affect boys at 2 or 3 years of age, and they eventually become wheelchair-bound by 12 years of age. Patients die from respiratory or cardiovascular complications in their late teens or twenties [1,2,3,4]. While all DMD patients have inexorably progressive cardiomyopathy (CMP), the age of onset and clinical progression of cardiac complications vary. Nevertheless, early diagnosis and treatment of CMP in DMD patients has been shown to improve both quality and length of life [1].

Transthoracic echocardiography (TTE) and cardiac magnetic resonance imaging (CMR) are imaging modalities with distinct advantages in demonstrating cardiac function in the DMD population [5]. Although many new developments in imaging technology and data processing provide important advances in the diagnosis of DMD CMP, studies on early and easy diagnosis of CMP due to DMD continue [5,6].

Arrhythmias may develop in the course of DMD CMP, and especially ventricular arrhythmias may be life-threatening. Because of the risk of arrhythmias and sudden cardiac death (SCD) that occurs in DMD, it is recommended that clinicians consider Holter monitors in patients with cardiac dysfunction [7]. However, studies support an increased risk of arrhythmia in DMD patients even before CMP has developed [8,9]. Nevertheless, identifying patients at risk for life-threatening arrhythmias has been difficult for many years. Given these concerns, screening and early detection of cardiac arrhythmias in DMD remains an understudied area. 

Premature ventricular contractions (PVCs) are a common arrhythmia that can cause symptoms of varying severity. PVCs have been associated with impaired ventricular function, known as PVC-induced CMP [10]. It has been shown that suppression of frequent PVCs may be associated with improved left ventricular (LV) function, both in the setting of PVC-induced and in the presence of other cardiomyopathies [10,11]. Moreover, frequent PVCs may lead to an increased risk of mortality in patients with structural heart disease [12]. It has been shown in previous studies that the frequency of PVC in DMD patients increases as CMP develops, and that the presence of PVC is associated with increased mortality [9].

The standard 12-lead electrocardiogram (ECG) remains the most frequently recorded non-invasive test in medicine due to its accessibility and simplicity. The Tpeak–Tend interval (Tp-e), one of the ECG parameters and an indicator of myocardial repolarization, has been shown to be associated with increased risk of malignant ventricular arrhythmias and SCD [13,14,15]. The most represented populations in these studies were long QT syndrome, short QT syndrome, Brugada syndrome, and other cardiomyopathies [16,17,18].

Evaluation of the duration of the repolarization period may be useful in the evaluation of increased arrhythmogenesis risk and in the early detection of arrhythmias in DMD. This simple, easily accessible, inexpensive, and non-invasive ECG parameter may be a useful indicator of arrhythmia in patients with PVC associated with LV dysfunction. The identification of risk groups will ensure the implementation of therapeutic and preventive measures together with lifestyle changes and pharmacotherapy. 

In this study, we investigated whether Tp-e interval prolongation has a role in predicting the frequency of PVC and cardiac dysfunction in patients with DMD.

## 2. Materials and Methods

Twenty-five patients with DMD whose cardiac functions were followed up in the Department of Pediatric Cardiology of Istanbul Medical Faculty were included in the study. TTE, ECG, and 24 h Holter monitoring were performed. Troponin and pro-BNP, which are biomarkers indicating heart failure (HF), were evaluated. Those with a frequency of <1% PVC in the 24 h Holter were assigned to Group 1 (*n* = 15), and those with >1% were assigned to Group 2 (*n* = 10). Comparisons were made with 27 healthy boys of similar age and body mass index (BMI). The study complied with the principles outlined in the Declaration of Helsinki. An informed consent form containing information about cardiac evaluation and examinations to be performed was obtained from the participants and their parents if they were younger than 18 years of age. The Local Ethics Committee approved the study (dated 19 November 2021 and numbered 622662).

All patients underwent an echocardiographic evaluation in accordance with recommendations of the American Society of Echocardiography [19]. The examinations were performed with an echocardiographic machine (GE Vivid 5) to acquire M-mode, 2-D mode, and Doppler images of LV dimensions, wall thickness, and function. Parasternal and apical views were obtained with the patient in the left lateral decubitus position. A modified Simpson’s method was used to assess the LV ejection fraction (LVEF). 

The ECGs were recorded at 25 mm/s with an amplitude of 10 mm/mV. P wave duration, PR interval, QRS duration, and QT interval were calculated and corrected for heart rate. The mean value of the three QT intervals was calculated for each lead. The QT interval was measured from the beginning of the QRS complex to the end of the T wave and corrected for heart rate using Bazett’s formula. The Tp-e interval (ms) was calculated with the tangent method [16]. The time from the peak of the T wave (or nadir if a negative or biphasic T wave was obtained) to the intersection between the tangent at the steepest point of the T wave and the isoelectric line was measured digitally in milliseconds with Cardio Calipers Version 3.3 software (Iconico, Inc., New York, NY, USA). The Tp–e interval was measured from the best available T wave in lead DII. Precordial leads V5 was used when DII was not suitable for analysis. Tp-e interval measurement was not performed from leads with T waves with amplitudes less than 1.5 mm. The 12-lead ECG variables were examined by an independent observer (HO), who was blinded to the clinical data. The recorded Tp-e value was the greatest value obtained by measuring two times by the observer.

Twenty-four hours of Holter monitoring was obtained by using a three-channel device. After computerized primary analysis, the records were edited manually. Minimum, maximum, and mean heart rates and PVC burden were evaluated during the 24 h follow-up. PVC burden was assessed as percentage and number of PVCs per day. The percentage of PVCs was determined by dividing the total number of PVCs by the total number of beats recorded during Holter monitoring. 

### Statistical Analysis

The Kolmogorov–Smirnov test was used to analyze the normality of the data. Parametric continuous data were expressed as mean ± standard deviation (SD), and non-parametric continuous data, median (minimum–maximum), and categorical data as percentages. A Chi-square test was used to assess the differences in categorical variables between the groups. ANOVA analysis was performed to compare all reported data for parametric variables, whereas the Kruskal–Wallis test was used for comparison among non-parametric variables between groups. The relationships among the parameters were assessed using Pearson’s or Spearman’s correlation analysis according to the normality of the data. Logistic regression analysis was used to determine independent predictors for PVC >1%. A receiver operating characteristic curve (ROC) was obtained to determine the best cut-off value of Tp-e in the prediction of PVC 1%. Significance was assumed at a two-sided *p* < 0.05. All statistical tests were conducted using the Statistical Package for the Social Sciences 26.0 for Windows (SPSS Inc., Chicago, IL, USA). 

## 3. Results

Our study included 25 DMD patients with a mean age of 12.94 ± 4.5 years (2–18 range) and 27 healthy children with a mean age of 13.2 ± 5 years with similar demographic characteristics. Among the DMD patients, there were 15 patients (mean age 11.7 ± 4.5 years) in Group 1 with PVC < 1% and 10 patients (mean age 14.6 ± 4.08 years) in Group 2 with PVC > 1%. 

In the TTE evaluation, LVEF was lowest in Group 2 and highest in the control group, and the differences were statistically significant between the three groups (50.5 ± 14.4, 59.9 ± 8.9, and 67.6 ± 2.7, respectively; *p* < 0.001). LV end-diastolic diameter (LVEDD) was greater in Group 2 than in Group 1 and the control group (50.4 ± 8.9, 43.7 ± 6.2, and 41.0 ± 6.3, respectively; *p* = 0.005). Interventricular septal thickness (IVSd) and posterior wall thickness (PWd) in diastole was thinner in Group 1 and Group 2 compared to the control group (*p* < 0.001 in both).

Pro-BNP and hs-troponin levels were higher in Groups 1 and 2 than in the control group (*p* = 0.001 and *p* < 0.001, respectively). 

In the ECG, it was observed that the PR interval was shorter and the QTc interval was longer in Groups 1 and 2 compared to the control group (*p* < 0.001 in both). The Tp-e interval was longer in Group 2 compared to Group 1 and the control group (80.7 ± 18.9, 68.8 ± 7.8, and 62.7 ± 2.8 ms, respectively; *p* < 0.001). 

Average heart rate (HR) was higher in Groups 1 and 2 compared to the control group (*p* = 0.016) during 24 h Holter monitoring. The burden of PVCs was higher in Group 2 than in Group 1 and the control group (5166.8 (1050–19,500), 218.2 (0–917), and 87.7 (0–1213), respectively; *p* < 0.001). While ventricular couplet beats were not observed in the control group and Group 1, an average of 5.4 ± 8.8 beats was observed in Group 2 (*p* < 0.001). The echocardiographic, laboratory, electrocardiographic, and Holter monitoring findings of the study groups are shown in Table 1.

The Tp-e interval was negatively correlated with LVEF (*r* = –0.312, *p* = 0.024) and positively correlated with pro-BNP (*r* = 0.571, *p* < 0.001), troponin (*r* = 0.446, *p* = 0.007), QTc interval (*r* = 0.432, *p* = 0.001), and PVC (*r* = 0.400, *p* = 0.003). PVC burden was negatively correlated with LVEF (*r* = –0.346, *p* = 0.012) and positively correlated with LVEDD (*r* = 0.407, *p* = 0.003), pro-BNP (*r* = 0.353, *p* = 0.038), troponin (*r* = 0.623, *p* < 0.001), and QTc interval (*r* = 0.459, *p* < 0.001). Correlations of Tp-e interval and PVC with echocardiographic, electrocardiographic, and laboratory parameters are shown in Table 2.

The parameters affecting PVC/24 h frequency of >1% were evaluated in a logistic regression analysis along with univariate and multivariate analyses. LVEF, LVEDD, Tp-e interval, QTc, and hs-troponin levels, which were statistically significant in the univariate analysis, were then reevaluated in a multivariate analysis. The LVEF (OR 0.879, 95% CI 0.812–0.953; *p* = 0.002) and Tp-e interval (OR 1.181, 95% CI 1.047–1.332; *p* = 0.007) were independent predictors of PVC/24 h frequency of >1% via multivariate logistic regression analysis (Table 3). 

In ROC curve analysis, a Tp-e interval > 71.65 ms predicted PVC >1%, with a sensitivity of 80% and a specificity of 90% (AUC = 0.842, 95% CI 0.663–1.000, *p* = 0.001) (Figure 1).

## 4. Discussion

In our study, we evaluated the relationship between the Tp-e interval and the frequency of PVCs in DMD patients because ventricular arrhythmias and PVCs, which are the most common among them, are common in patients with ventricular dysfunction and are an early indicator of progressive myocardial decline.

According to the results of our study, DMD Group 2 patients with PVC > 1% had higher LVEDD, pro-BNP, and hs-troponin values than DMD Group 1 and the control group, while LVEF was found to be lower. LV dysfunction was observed together with arrhythmia in Group 2. QTc and Tp-e intervals were significantly higher in Group 2 compared to Group 1 and the control group. The Tp-e interval was negatively correlated with LVEF and positively correlated with PVC. The LVEF and Tp-e interval were independent predictors of PVC/24 h frequency of >1%. To the best of our knowledge, this is the first study to evaluate the potential of the Tp-e interval on the surface ECG to play a simple screening tool role in determining cardiac status in patients with DMD.

Dystrophin is an important giant molecule linking the cytoskeleton to the plasma membrane in muscle cells and is located on the cytoplasmic side of the plasma membrane of muscle fibers, transmitting force and concomitantly protecting the plasmalemma [1]. Dystrophin deficiency in the heart leads to a loss of sarcolemma integrity, triggering muscle degradation followed by necrosis, fibrosis, and fibro-fatty replacement of normal cardiac muscle tissue, commonly resulting in lethal CMP, though the onset and progression of this phenotype vary [2,3,4]. Clinically significant CMP typically occurs in the middle of the second decade [20,21]. However, pre-clinical cardiac involvement is thought to be present in DMD patients without the development of dilated CMP [22,23].

Due to the underlying pathogenesis, the dilated CMP of DMD is associated with rhythm abnormalities, particularly conduction defects, bradycardia, and ventricular arrhythmias [24,25]. ECG abnormalities can be detected in up to 60% of DMD patients, and these changes are independent of the presence of CMP. The most common short PR interval was defined, and a prolonged QTc interval was reported [26,27]. Arrhythmias when LV function deteriorates are associated with higher-grade arrhythmias [7]. However, studies show that patients with normal LVEF may also have conduction system involvement [28,29], and that patients are at risk for ventricular arrhythmias and SCD even before DMD CMP develops [8,9].

In a study investigating the prevalence and prognostic value of PVC with 24 h Holter monitoring in 45 DMD patients who did not have congestive HF yet, the patients were followed for three years. Patients who died suddenly were more likely to have PVC at the initial 24 h Holter monitoring. As a result of the study, they emphasized that PVCs are often associated with asymptomatic LV dysfunction, and complex ones are one of the risk factors for SCD [8].

In another study investigating the prevalence and prognostic significance of ventricular arrhythmias in DMD, 80 patients were followed for 5 years. PVCs were found to be 30% at baseline, and the incidence of PVC increased as the clinical severity of skeletal muscle involvement progressed. In a 5-year period, four of the 33 deaths were sudden. Complex PVCs were observed in three of these four patients. They showed that ventricular arrhythmias are a common complication of DMD, and the incidence of ventricular arrhythmias increases with progression of myocardial involvement [9]. Similarly, in our study, LV functions were worse in the group with frequent PVCs. Screening and prediction of PVCs may contribute to prognosis and LV function in these patients, even if they are asymptomatic. 

However, ventricular arrhythmia and SCD in DMD patients are difficult to predict and prevent, and the underlying mechanisms are unclear.

There are studies showing that CMR, which detects myocardial scarring and late gadolinium increase, independently predicts mortality, cardiovascular events, and ventricular arrhythmias in those with mild LV dysfunction or even preserved LV function [5,30]. However, CMR is often not a common modality, both because of its cost and the progressive skeletal deformity in patients with DMD. Moreover, the standard 12-lead ECG is readily available and requires minimal technical expertise and remains the most frequently recorded non-invasive test.

There has been great interest in the relationship between arrhythmic risk and repolarization patterns and their ECGs correlate. Increased frequency of PVCs has been shown to mean greater enlargement of the impaired myocardium and hence worse LV systolic function. Therefore, it is reasonable to speculate that greater global ventricular repolarization abnormalities reflected by increased ECG repolarization parameters may parallel more frequent PVCs and thus LV dysfunction. Several myocardial repolarization markers, such as Tp–Te, QT and corrected QT (QTc) intervals, QT distribution (QTd), Q-Tpeak (QTp), and Tp-Te/QTc ratio, which can be obtained from ECGs, have been reported to be associated with ventricular arrhythmic risk and furthermore with cardiovascular mortality and morbidity [31,32]. 

The potential of the Tp-e range to serve as a risk marker for the development of cardiac arrhythmias has been demonstrated by several studies in different populations. In previous studies, Tp-e was associated with torsades de pointes and SCD in patients with Brugada syndrome, long QT syndrome, hypertrophic cardiomyopathy, and myocardial infarction [16,18,33]. In patients with LV systolic dysfunction and an implanted ICD, Tp-e independently predicted ventricular tachyarrhythmia, appropriate ICD therapy, and overall mortality [34,35]. In recent studies, Tp-e interval prolongation was associated with higher PVC burden in patients with structurally normal hearts, and the Tp-e interval has been shown to be helpful in predicting sustained idiopathic VT in patients with idiopathic PVC [36,37]. Since one of the re-entry requirements is the local distribution of myocardial repolarization, it is reasonable to expect that an increase in the total ventricular distribution of repolarization (DVR) may predispose patients to ventricular tachyarrhythmias and cardiac arrest.

The Tp-e interval of the T wave corresponds to transmural repolarization distribution (TDR) in the ventricular myocardium [17,38]. Studies showed that Tp-e may reflect more accurately the total DVR [34]. This is a period when the epicardium is repolarized and fully excitable, but M cells in the subendocardium are still in the process of repolarization and are susceptible to the possibility of early depolarizing activity [39]. When appropriate, initiation of a reentry circuit after a critical early depolarization can cause ventricular arrhythmia, ventricular tachycardia, or ventricular fibrillation [40,41]. 

There are limited data on the relationship of repolarization parameters with LV function beyond being a marker of arrhythmic risk. The relationship between repolarization parameters on ECG and LV systolic function in Chagas disease was investigated. All repolarization parameters, including Tp-e, were significantly elevated in patients with moderate or severe LV systolic dysfunction. Frequent PVC presence and the QT distribution, another marker of repolarization, were the best markers for asymptomatic LV dysfunction [42].

Although the CMP of DMD is a very suitable candidate for the evaluation of ventricular repolarization parameters due to its main features such as myocardial fibrosis, fat infiltration, ventricular dilatation, increased arrhythmia, and SCD risk, the Tp-e interval has not yet been studied in these patients.

The main mechanism of Tp-e interval prolongation and the ventricular repolarization anomaly is the ion transport problem of cells in different layers of the ventricular myocardium [43]. There are studies showing the role of ion channels in arrhythmia mechanisms in DMD patients [44]. It has been suggested that changes in the functional expression of Na and K ion channels associated with a specific mutation in the dystrophin gene lead to QRS widening and QTc prolongation that can be seen in patients with DMD and play a role in the formation of ventricular arrhythmias. In our study, QTc was significantly prolonged in the DMD group with an increased frequency of PVC [45]. 

Other studies emphasize that abnormalities of cardiac L-type calcium channels in dystrophic cardiomyocytes may lead to arrhythmias by disrupting the electrophysiology of the heart [46]. During an action potential (AP) in dystrophic mouse cardiomyocytes, calcium influx was shown to increase and AP to be prolonged. Significant shortening of the PQ intervals and a slight prolongation of the QT intervals were detected in the ECGs of these mice. Since calcium is the main depolarizing current during early repolarization, prolonged AP during this phase may explain the repolarization abnormality such as QTc prolongation that can be observed in DMD patients.

Histological changes in the heart of patients with DMD include fibrosis, degeneration, and fat infiltration [47]. In addition, these myocardial pathologies can also lead to heterogeneous myocardial electrical activation and increased DVR. It seems that cardiac conduction and repolarization anomalies are inherent features of DMD. Regardless of the molecular and/or structural mechanisms underlying the increased repolarization distribution, our results support the hypotheses that a repolarization abnormality exists in DMD and that Tp-e prolongation is a useful marker of arrhythmia risk in DMD patients. Tp-e can be valuable for risk stratification and help guide treatment decisions.

Early detection and management of CMP associated with DMD and the arrhythmias that may develop in this process are critical because cardioprotective medical treatments can slow or reverse remodeling and reduce HF symptoms in these patients. Observing a prolongation in the Tp-e interval may enable early diagnosis of the risk of developing HF and arrhythmia.

Our study has some limitations. Most importantly, because it was a single center study and DMD is a rare disease, the sample was small. In addition, long-term follow-up of the patients was not available. There is a need for larger studies showing the effect of the Tp-e interval on long-term prognosis and mortality.

## 5. Conclusions

DMD is associated with dilated CMP and rhythm abnormalities, primarily arrhythmias. Dilated CMP seen in these patients is characterized by extensive fibrosis of the LV free wall. As the disease progresses, HF and arrhythmias eventually develop. The onset of heart injury in children with DMD is inconspicuous, and the prognosis is poor once it develops to the stage of HF. Therefore, early diagnosis and treatment of cardiovascular disease are critical to the survival and/or quality of life of these patients. Imaging methods are essential in the evaluation of cardiac functions in patients with DMD, but determining Tp-e prolongation from ECG data may be useful for easy and early diagnosis of the course of the disease.

## Figures and Tables

**Figure 1 diagnostics-13-02381-f001:**
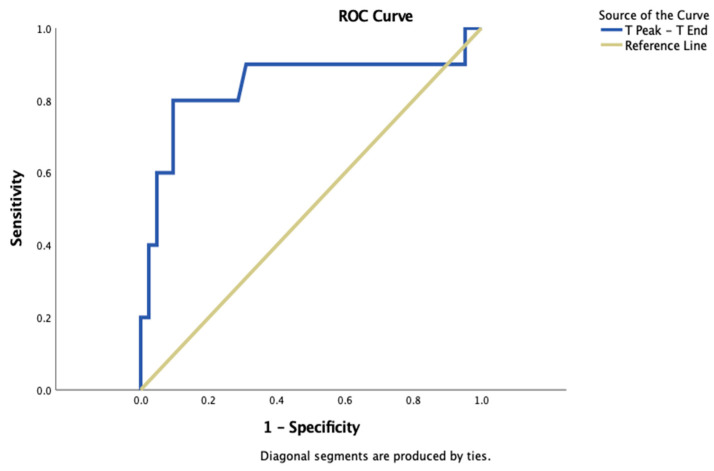
ROC curve analysis showing the specificity and sensitivity of the Tp-e in predicting PVC > 1%.

**Table 1 diagnostics-13-02381-t001:** Clinical, echocardiographic, electrocardiographic, and Holter monitoring findings of study groups.

	Control Group (*n* = 27)	DMD, PVC < 1% Group 1 (*n* = 15)	DMD, PVC > 1% Group 2 (*n* = 10)	*p*-Value
Age (years)	13.2 ± 5	11.7 ± 4.5	14.6 ± 4.08	0.214 ^K^
BMI (kg/m^2^)	25.7 ± 3.3	26.3 ± 6.6	26.9 ± 6.8	0.427 ^K^
Echocardiography				
LVEF (%)	67.6 ± 2.7 ^x,y^	59.9 ± 8.9 ^x,z^	50.5 ± 14.4 ^y,z^	<0.001 ^A^
LVEDD (mm)	41.0 ± 6.3 ^y^	43.7 ± 6.2 ^z^	50.4 ± 8.9 ^y,z^	0.005 ^A^
IVSd (mm)	8.5 ± 1.2 ^x,y^	6.8 ± 0.8 ^x^	6.5 ± 0.6 ^y^	<0.001 ^K^
PWd (mm)	8.6 ± 1.1 ^x,y^	6.7 ± 0.7 ^x^	6.8 ± 0.5 ^y^	<0.001 ^K^
Cardiac biomarkers				
Pro-BNP (pg/mL)	449 ± 18.3 ^x,y^	197.3 ± 235.1 ^x^	437.3 ± 574.8 ^y^	0.001 ^K^
Hs-Troponin (pg/mL)	3.1 ± 1 ^x,y^	62.9 ± 51.2 ^x^	98.2 ± 32.1 ^y^	<0.001 ^K^
Electrocardiography				
PR interval (ms)	140.9 ± 17.6 ^x,y^	117.5 ± 14.7 ^x^	124.2 ± 19.4 ^y^	<0.001 ^K^
QRS duration (ms)	95.3 ± 12.7	87.3 ± 5.7	94.4 ± 7.1	0.067 ^K^
QTc interval (ms)	396.2 ± 14.9 ^x,y^	415.0 ± 19.9 ^x^	431.0 ± 20.4 ^y^	<0.001 ^K^
Tp-e interval (ms)	62.7 ± 2.8 ^y^	68.8 ± 7.8 ^z^	80.7 ± 18.9 ^y,z^	<0.001 ^K^
24 h Holter monitoring				
Max HR (bpm)	159.0 ± 21.2	156.5 ± 12.7	149.0 ± 21.8	0.516 ^K^
Min HR (bpm)	51.3 ± 12.8	56.9 ± 6.5	55.4 ± 6.9	0.222 ^A^
Average HR (bpm)	84.3 ± 16.6 ^x,y^	91.9 ± 12.4 ^x^	92.9 ± 12.9 ^y^	0.016 ^K^
PVC per 24 h	87.7 (0–1213) ^y^	218.2 (0–917) ^z^	5166.8 (1050–19,500) ^y,z^	<0.001 ^K^
Couplets per 24 h	0 ± 0 ^y^	0 ± 0 ^z^	5.4 ± 8.8 ^y,z^	<0.001 ^K^
Triplets per 24 h	0 ± 0	0 ± 0	1 ± 1	

^A^ ANOVA (Tukey test/Tamhane)/^K^ Kruskal–Wallis (Mann–Whitney u test). ^x^: *p* < 0.05 between control group and Group 1, ^y^: *p* < 0.05 between control group and Group 2, ^z^: *p* < 0.05 between Group 1 and Group 2.

**Table 2 diagnostics-13-02381-t002:** Correlation of Tpeak–Tend and PVC with echocardiographic, electrocardiographic, and laboratory parameters.

Variables	Tpeak–Tend		PVC	
	*r*	*p*-Value	*r*	*p*-Value
LVEF %	−0.312	0.024 *	−0.346	0.012 *
LVEDD (mm)	0.115	0.415	0.407	0.003 *
Pro-BNP	0.571	<0.001 *	0.353	0.038 *
Hs-Troponin	0.446	0.007 *	0.623	<0.001 *
PR interval (ms)	−0.247	0.077	–0.083	0.561
QRS duration (ms)	−0.224	0.110	0.106	0.453
QTc interval (ms)	0.432	0.001 *	0.459	<0.001 *
Tp-Te	—	—	0.400	0.003 *
PVC	0.400	0.003 *	—	—
Maximum HR	0.002	0.990	–0.102	0.471
Minimum HR	0.266	0.057	0.170	0.229
Average HR	0.267	0.056	0.245	0.080

* *p* < 0.05.

**Table 3 diagnostics-13-02381-t003:** Univariate and multivariate regression analysis of risk factors associated with PVC > 1%.

Univariate Regression						Multivariate Regression				
	OR	%95 CI			*p*-Value	OR	%95 CI			*p*-Value
LVEF %	0.879	0.812	-	0.953	0.002 *	0.879	0.812	-	0.953	0.002 *
Tp-e	1.195	1.068	-	1.336	0.002 *	1.181	1.047	-	1.332	0.007 *
QTc	1.088	1.027	-	1.153	0.004 *					
LVEDD	1.210	1.040	-	1.408	0.014 *					
Pro-BNP	1.002	1.000	-	1.005	0.074					
Hs-Troponin	1.025	1.006	-	1.044	0.010 *					
PR interval	0.977	0.939	-	1.016	0.240					
QRS duration	1.017	0.954	-	1.085	0.603					

* *p* < 0.05.

## Data Availability

Data supporting the reported results are available from the principal author, S.B., upon request.

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
