# Peer review of "Prognostic Value of Tpeak–Tend Interval in Early Diagnosis of Duchenne Muscular Dystrophy Cardiomyopathy"

_diagnostics, 2023, doi:10.3390/diagnostics13142381_

Round 1

Reviewer 1 Report

29 1. Introduction 30

Duchenne muscular dystrophy (DMD) is a severe, progressive, X-linked genetic 31 disease that affects roughly one in every 5,000 live male births. This progressive disease 32 causes severe clinical symptoms; progressive muscle weakness typically begins to affect 33 boys at 2 or 3 years of age, and they eventually become wheelchair-bound by 12 years of 34 age. Patients die from respiratory or cardiovascular complications in their late teens or 35 twenties [1,2,3,4]. While all DMD patients have inexorably progressive cardiomyopathy 36 (CMP), the age of onset and clinical progression of cardiac complications vary. Never- 37 theless, early diagnosis and treatment of CMP in DMD patients has been shown to im- 38 prove both quality and length of life [1]. 39

Transthoracic echocardiography (TTE) and cardiac magnetic resonance imaging 40 (CMR) are imaging modalities with distinct advantages in demonstrating cardiac func- 41 tion in the DMD population [5]. Although many new developments in imaging 42 technology and data processing provide important advances in the diagnosis of DMD 43 CMP, studies on early and easy diagnosis of CMP due to DMD continue [5,6]. 44

Arrhythmias may develop in the course of DMD CMP, and especially ventricular 45 arrhythmias may be life-threatening. Because of the risk of arrhythmias and sudden car- 46 diac death (SCD) that occur in DMD, it is recommended that clinicians consider Holter 47 monitors in patients with cardiac dysfunction [7]. Arrhythmias and conduction disease 48 occur after the development of CMP [8]. However, studies support an increased risk of 49 arrhythmia in DMD patients even before CMP has developed [8,9]. Given these 50 concerns, screening for cardiac arrhythmias in DMD remains an understudied area. 51

Premature ventricular contractions (PVCs) are a common arrhythmia that can 52 cause symptoms of varying severity. PVCs have been associated with impaired 53 ventricular function, known as PVC-induced CMP [10]. It has been shown that 54 suppression of frequent PVCs may be associated with improved left ventricular (LV) 55 function, both in the setting of PVC-induced and in the presence of other 56 cardiomyopathies [10,11]. Moreover, frequent PVCs may lead to an increased risk of 57 mortality in patients with structural heart disease [12]. It has been shown in previous 58 studies that the frequency of PVC in DMD patients increases as CMP develops, and that 59 the presence of PVC is associated with mortality [9]. 60

Tpeak-Tend interval (Tp-e), one of the electrocardiography (ECG) parameters and 61 an indicator of myocardial re-polarization, is a predictor of ventricular tachyarrhythmia 62 and has been associated with cardiovascular mortality and morbidity [13,14,15]. 63

This simple, easily accessible, inexpensive and non-invasive ECG parameter may 64 be useful indice of arrhythmia in patients with PVC associated with LV dysfunction. 65

In this study, we aimed to detect cardiac dysfunction early and easily in patients 66 with DMD. We investigated whether the prolongation of the Tp-e interval on ECG plays 67 a role in predicting the frequency of PVCs and early cardiac impairment in these pa- 68 tients. 69

2. Materials and Methods 70

Twenty-five patients with DMD whose cardiac functions were followed up in the 71 Department of Pediatric Cardiology of Istanbul Medical Faculty were included in the 72 study. TTE, ECG, and 24-hour Holter monitoring were performed. Troponin and pro- 73 BNP, which are biomarkers indicating heart failure (HF), were evaluated. Those with a 74 frequency of < 1% PVC in the 24-hour Holter were assigned to Group 1 (n = 15), and 75 those with > 1% were assigned to Group 2 (n = 10). Comparisons were made with 27 76 healthy boys of similar age and body mass index (BMI). The study complied with the 77 principles outlined in the Declaration of Helsinki. An informed consent form was ob- 78 tained from all participants. The Local Ethics Committee approved the study (dated 79 19/11/2021 and numbered 622662). 80

All patients underwent an echocardiographic evaluation in accordance with rec- 81 ommendations of the American Society of Echocardiography [16]. The examinations 82 were performed with an echocardiographic machine (GE Vivid 5) to acquire M-mode, 2- 83 D mode, and Doppler images of LV dimensions, wall thickness, and function. Paraster- 84 nal and apical views were obtained with the patient in the left lateral decubitus position. 85 A modified Simpson's method was used to assess the LV ejection fraction (LVEF). 86

The ECGs were recorded at 25 mm/s with an amplitude of 10 mm/mV. P wave du- 87 ration, PR interval, QRS duration, and QT interval were calculated and corrected for 88 heart rate. The mean value of the three QT intervals was calculated for each lead. The 89 QT interval was measured from the beginning of the QRS complex to the end of the T 90 wave and corrected for heart rate using Bazett’s formula. For Tp-e interval measure- 91 ment, the tangent method was used in leads V5 and DII. Tp-e interval measurement was 92 not performed from leads with T waves with amplitudes less than 1.5 mm. The time 93 from the peak of the T wave to the intersection between the tangent at the steepest point 94 of the T wave and the isoelectric line was measured digitally in milliseconds with Cardio 95 Calipers Version 3.3 software (Iconico, Inc., New York, NY, USA). The 12-lead ECG vari- 96 ables were examined by an independent observer (HO), who was blinded to the clinical 97 data. 98

Twenty‐four hours of Holter monitoring was obtained by using a three-channel 99 device. After computerized primary analysis, the records were edited manually. Mini- 100 mum, maximum, mean heart rates, and PVC burden were evaluated during the 24-hour 101 follow-up. PVC burden was assessed as percentage and number of PVCs per day. The 102 percentage of PVCs was determined by dividing the total number of PVCs by the total 103 number of beats recorded during Holter monitoring. 104

2.1. Statistical analysis 105

The Kolmogorov–Smirnov test was used to analyze the normality of the data. 106 Parametric continuous data are expressed as mean ± standard deviation (SD); non- 107 parametric continuous data, median (minimum-maximum) and categorical data as 108 percentages. A Chi-square test was used to assess the differences in categorical variables 109 between the groups. The ANOVA analysis was performed to compare all reported data 110 for parametric variables, whereas the Kruskal–Wallis test was used for comparison 111 among non-parametric variables between groups. The relationships among the 112 parameters were assessed using Pearson’s or Spearman’s correlation analysis according 113 to the normality of the data. Logistic regression analysis was used to determine 114 independent predictors for PVC >1%. The receiver operating characteristic curve (ROC) 115 curve was obtained to determine the best cut-off value of Tp-e in the prediction of PVC 116 1%. Significance was assumed at a two-sided p < 0.05. All statistical tests were conducted 117 using the Statistical Package for the Social Sciences 26.0 for Windows (SPSS Inc., 118 Chicago, IL, USA). 119

120

3. Results 121

Our study included 25 DMD patients with a mean age of 12.94 ± 4.5 years (2–18 range) 122 and 27 healthy children with a mean age of 13.2 ± 5 years with similar demographic char- 123 acteristics. Among the DMD patients, there were 15 patients (mean age 11.7 ± 4.5 years) 124 in Group 1 with PVC < 1% and 10 patients (mean age 14.6 ± 4.08 years) in Group 2 with 125 PVC > 1%. 126

In the TTE evaluation, LVEF was lowest in Group 2 and highest in the control group, 127 and the differences were statistically significant between the three groups (50.5 ± 14.4, 59.9 128 ± 8.9, and 67.6 ± 2.7, respectively, p < 0.001). LV end-diastolic diameter (LVEDD) was 129 greater in Group 2 than in Group 1 and the control group (50.4 ± 8.9, 43.7 ± 6.2, and 41.0 ± 130 6.3, respectively, p = 0.005). Interventricular septal thickness (IVSd) and posterior wall 131 thickness (PWd) in diastole was thinner in Group 1 and Group 2 compared to control 132 group (p < 0.001 in both). 133

Pro-BNP and hs-troponin levels were higher in Groups 1 and 2 than in the control 134 group (p = 0.001 and p < 0.001, respectively). 135 In the ECG, it was observed that the PR interval was shorter and the QTc interval 136 was longer in Groups 1 and 2 compared to the control group (p < 0.001 in both). Tp-e 137 interval was longer in Group 2 compared to Group 1 and the control group (80.7 ± 18.9, 138 68.8 ± 7.8, and 62.7 ± 2.8 ms, respectively, p < 0.001). 139

Average heart rate (HR) was higher in Groups 1 and 2 compared to the control group 140 (p = 0.016) during 24-hour Holter monitoring. The burden of PVCs was higher in Group 2 141 than in Group 1 and the control group [5166.8 (1050–19500), 218.2 (0–917), and 87.7 (0– 142 1213), respectively, p < 0.001]. While ventricular couplet beats were not observed in the 143 control group and Group 1, an average of 5.4 ± 8.8 beats was observed in Group 2 (p < 144 0.001). The echocardiographic, laboratory, electrocardiographic, and Holter monitoring 145 findings of the study groups are shown in Table 1. 146

     170 The parameters affecting PVC/24-hour frequency of > 1% were evaluated in a logistic 171 regression analysis along with univariate and multivariate analyses. LVEF, LVEDD, Tp-e 172 interval, QTc, and hs-troponin levels, which were statistically significant in the univariate 173 analysis, were then reevaluated in a multivariate analysis. The LVEF (OR 0.879, 95% CI 174 0.812–0.953; p = 0.002) and Tp-e interval (OR 1.181, 95% CI 1.047–1.332; p = 0.007) were 175 independent predictors of PVC/24-hour frequency of > 1% via multivariate logistic regres- 176 sion analysis (Table 3). 177 178 Table 3: Univariate and multivariate regression analysis of risk factors associated 179

188 4. Discussion 189

Dystrophin is an important giant molecule linking the cytoskeleton to the plasma 190 membrane in muscle cells and is located on the cytoplasmic side of the plasma mem- 191 brane of muscle fibers, transmitting force and concomitantly protecting the plasma- 192 lemma [1]. Dystrophin deficiency in the heart leads to a loss of sarcolemma integrity, 193 triggering muscle degradation followed by necrosis, fibrosis, and fibro-fatty replacement 194 of normal cardiac muscle tissue, commonly resulting in lethal CMP, though the onset 195 and progression of this phenotype vary [2,3,4]. Clinically significant CMP typically oc- 196 curs in the middle of the second decade. Studies have suggested that the average age for 197 the development of abnormal LVEF is 14.3 years [17]. Eventually, most patients with 198 DMD develop cardiomyopathy by 20 years of age [18]. However, pre-clinical cardiac 199 involvement is thought to be present in up to one-fourth of DMD patients under 6 years 200 old [19,20]. 201

The dilated CMP seen in these patients is characterized by widespread fibrosis of 202 the LV free wall. Therefore, dilated CMP of DMD is associated with rhythm abnormali- 203 ties, especially ventricular arrhythmias [21,22,23]. Due to the underlying pathogenesis 204 (myocyte disruption attributable to abnormal dystrophin protein), conduction system 205

182 involvement may also be seen in patients with normal LVEF [22,24,25]. Many will be at a 206 high risk for arrhythmia and SCD, which contributes considerably to the morbidity and 207 mortality of the disease [19,26]. However, the diagnosis and prevention of arrhythmia 208 are challenging in DMD patients. 209

ECG abnormalities can be detected in up to 60% of DMD patients, and these in- 210 clude conduction defects, bradycardia, ventricular arrhythmias, and sudden death 211 [19,22,23]. There are studies showing the role of sodium and potassium ion channels in 212 controlling cardiac excitability in arrhythmia mechanisms in DMD patients [27]. Ar- 213 rhythmias seen when LV function worsens are associated with higher-grade arrhyth- 214 mias [7]. However, studies show that patients are at risk for ventricular arrhythmias 215 even before DMD CMP develops [8,9]. 216

In a study investigating the prevalence and prognostic value of PVC with 24-h 217 Holter monitoring in 45 DMD patients who did not have congestive HF yet, the patients 218 were followed for three years. Patients who died suddenly were more likely to have 219 PVC at the initial 24-h Holter monitoring. As a result of the study, they emphasized that 220 PVCs are often associated with asymptomatic LV dysfunction and complex ones are one 221 of the risk factors for SCD [8]. 222

In another study to investigate the prevalence and prognostic significance of 223 ventricular arrhythmias in DMD, 80 patients were followed for 5 years. PVCs were 224 found to be 30% at baseline, and the incidence of PVC increased as the clinical severity 225 of skeletal muscle involvement progressed. In a 5-year period, four of the 33 deaths were 226 sudden. Complex PVCs were observed in three of these four patients. They showed that 227 ventricular arrhythmias are a common complication of DMD, and the incidence of 228 ventricular arrhythmias increases with progression of myocardial involvement [9]. 229 Similarly, in our study, LV functions were worse in the group with frequent PVCs. 230 Screening and prediction of PVCs may contribute to prognosis and LV function in these 231 patients, even if they are asymptomatic. 232

While TTE is the standard imaging modality for screening for cardiovascular in- 233 volvement, CMR can be used to detect myocardial scarring in the diagnosis of early car- 234 diac functional impairment in normal LV function. There are studies showing that late 235 gadolinium increase in CMR independently predicts mortality, cardiovascular events, 236 and ventricular arrhythmias even in those with mild LV dysfunction or even preserved 237 left ventricular function. This indicates additional prognostic value beyond echocardiog- 238 raphy [5,28]. However, CMR is not a suitable modality for patients with DMD due to its 239 progressive skeletal deformity and inability to tolerate recumbent immobility for the 240 required amount of time. 241

Meanwhile, standard 12-lead ECG is readily available and requires minimal 242 technical expertise and remains the most frequently recorded non-invasive test. 243

The Tp-e interval of the T wave corresponds to the transmural repolarization dis- 244 tribution in the ventricular myocardium [29]. This is a period when the epicardium is 245 repolarized and fully excitable but M cells in the subendocardium are still in the process 246 of repolarization and are susceptible to the possibility of early depolarizing activity [30]. 247 When appropriate, initiation of a reentry circuit after a critical early depolarization can 248 cause ventricular arrhythmia, ventricular tachycardia, or ventricular fibrillation [31]. The 249 potential of the Tp-e interval to serve as a risk marker for the development of cardiac 250 arrhythmias has been demonstrated by various studies in different populations [32,33]. 251

 In our study, we evaluated the relationship between the Tp-e interval and the fre- 252 quency of PVCs in DMD patients. Because ventricular arrhythmias and VVCs, which are 253 the most common among them, are common in patients with ventricular dysfunction 254 and are an early indicator of progressive myocardial decline. 255

According to the results of our study, DMD Group 2 patients with PVC >%1 had 256 higher LVEDD, pro-BNP, and hs-troponin values than Group 1 and the control group, 257 while LVEF was found to be lower. LV dysfunction was observed together with arrhyth- 258 mia in Group 2. QTc and Tp-e intervals were significantly higher in Group 2 compared 259 to Group 1 and the control group. The LVEF and Tp-e interval were independent predic- 260 tors of PVC/24-hour frequency of > 1%. 261

Early detection and management of CMP associated with DMD and the arrhyth- 262 mias that may develop in this process is critical because cardioprotective medical treat- 263 ments can slow or reverse remodeling and reduce HF symptoms in these patients. Ob- 264 serving a prolongation in the Tp-e interval may enable early diagnosis the risk of devel- 265 oping HF and arrhythmia. 266

Our study has some limitations. Most importantly, because it was a single center 267 study and DMD is a rare disease, the sample was small. In addition, long-term follow- 268 up of the patients was not available. There is a need for larger studies showing the effect 269 of Tp-e interval on long-term prognosis and mortality. 270

5. Conclusions 271

DMD is associated with dilated CMP and rhythm abnormalities, primarily arrhyth- 272 mias. Dilated CMP seen in these patients is characterized by extensive fibrosis of the LV 273 free wall. As the disease progresses, HF and arrhythmias eventually develop. The onset 274 of heart injury in children with DMD is inconspicuous, and the prognosis is poor once it 275 develops to the stage of HF. Therefore, early diagnosis and treatment of cardiovascular 276 disease are critical to the survival and/or quality of life of these patients. Imaging methods 277 are essential in the evaluation of cardiac functions in patients with DMD, but determining 278 Tp-e prolongation from ECG data may be useful for easy and early diagnosis of the course 279 of the disease. 280

Overall, a well-done study to study cardiac involvement in special population of DMD. I have the following suggestions:

Introduction:

The introduction provided a good overview of Duchenne muscular dystrophy (DMD), its implications on cardiac health, and the need for early detection and treatment. However, it could benefit from more clarity; the authors give an overview but the significance of ECG parameter being studied is barely touched upon. Why are we looking at that specific parameter? Has there been other ECG related parameters? The overview of DMD can be more concise and authors should discuss more on the significance and aim of the specific  study ie investigating the cardiac involvement in DMD patients.

Materials and Methods:

The materials and methods section is clearly written and provides a good overview of how the study was conducted. However, it could be improved by:

Line 78: Specify what the informed consent form entails and whether the participants' legal guardians were also required to sign given the young age of the subjects.

In lines 87-97, it would be helpful to provide a bit more detail on how ECG data was interpreted, and how they ensured consistency and accuracy in measuring these parameters.

Discussion; Please present the major findings of the paper in the first paragraph to direct the readers. Currently, the discussion gives a major overview/review of DMD and then discusses the major findings only at the end. There should be more discussion on EKG findings with Tp-e interval. What are the different populations that this has been utilized in the past and what is the clinical significance of it?

The discussion about the Tp-e interval and its potential as a risk marker for cardiac arrhythmias is interesting and could be the crux of a novel approach for early detection. However, more detailed information on how the Tp-e interval was measured and interpreted could improve clarity.

The correlation established between left ventricular ejection fraction (LVEF), Tp-e interval, and PVC frequency is an important finding of the study. However, the authors could benefit from discussing these findings within the broader context of other studies, if any, on this topic.

The discussion on early detection and management strategies is pertinent, but the authors might want to delve deeper into specific management strategies and how they could be tailored based on the findings of this study.

adequate

Author Response

Dear editor, associate editors and reviewers,

First of all thank you for your interest about the manuscript, ‘’Prognostic value of Tpeak-Tend interval in early diagnosis of Duchenne muscular dystrophy cardiomyopathy’’. We would like to let you know that we have made the corrections suggested by the reviewers and have marked them in red.

Response to Reviewer 1: Thank you very much for your contribution. We are thankful to you for your time and providing thoughtful comments to strengthen our manuscript.

  1. You are absolutely right in your criticisms. We sought to expand the introduction and discussion sections on Tp-e especially about the different populations in which Tp-e is used and its clinical significance, and to explain the mechanisms that may cause Tp-e prolongation in DMD. We discussed more about the importance and purpose of the study. In the discussion section, we first presented the study findings and rearranged the paragraphs.
  2. We have detailed Tp-e measurement and interpretation in the Method section. Intraobserver and interobserver variability was not measured. This is the limitation of our study.
  3. We added that the legal guardians of the participants should also sign the informed consent form when the age of the subjects is younger than 18.

Reviewer 2 Report

Thank you for allowing me to review this paper on measuring Tpeak-Tend in DMD patients.  I feel this is an interesting measurement to help determine cardiac risk in DMD patients. I do have a few questions.

1. In your abstract you state that Tp-e can be used for easy and early diagnosis of disease. I believe this is wrong as your data suggests that this is only prolonged after you have disease therefore not for early diagnosis.

2. Line 45 "Arrhythmias may develop in the course of DMD CMP , and especially ventricular arrhythmias may be life-threatening." This sentence needs to be re-worded.

3. Line 48-49 needs to say "may occur".

4. Line 60 should read the presence of PVC's is associated with "increased" mortality".

5. In your results the patients with <1% and >1% PVC's have a difference in age which with a progressive disease such as DMD would be expected. This suggests that PVC's are a secondary finding "after the damage is done." thus this would lead to helping see there is CMP but not diagnoising it.

6. Table 1- Tp-e in the DMD Group 2 has a very large variation compared to the control and the Group 1. With this large variation was it one outlier that skewed the data? This range goes over the norms mean for Group 1 making this measurement difficulty to state if it is the findings or if just having increased PVC's is the marker for having CMP in DMD patients.

7. Line 239-240 You state CMR is not a suitable modality. This is conjecture on your part as CMR is frequently used on these individuals. This should be removed or reworded to not be your opinion.

8. Line 253 it states VVC's- this should be changed to PVC's.

Author Response

Dear editor, associate editors and reviewers,

First of all thank you for your interest about the manuscript, ‘’Prognostic value of Tpeak-Tend interval in early diagnosis of Duchenne muscular dystrophy cardiomyopathy’’. We would like to let you know that we have made the corrections suggested by the reviewers and have marked them in red.

Yours’ faithfully

 Response to Reviewer 2: Thank you very much for your contribution. We are thankful to you for your time and providing thoughtful comments to strengthen our manuscript.

  1. You are extremely correct in your criticism. We used the term early diagnosis because it may predict the frequency of PVC even before EF impaired. It may also be useful in early detection of arrhythmia risk. We have rearranged the term in line with your suggestion.
  2. As you suggested, we reworded the sentence and paragraph.
  3. We reworded the paragraph and added ‘may’.
  4. We added ‘increased’.
  5. You are absolutely right. Actually, that's exactly what we wanted to explain. We have revised the sentences so that they do not have a different meaning.
  6. The variation in Tp-e in group 2 was higher than in group 1 and controls. Significantly higher and similar values were found together in this group compared to the other groups, and the small number of patients is one of the reasons.
  7. We tried to discuss about the difficulties of widespread use of CMR due to both its expensive technique and the skeletal deformities of DMD patients. We rearranged the paragraph in line with your suggestion.
  8. We changed VVC to PVC.

Round 2

Reviewer 1 Report

The comments have been addressed adequately.